# Procrastinating with Confidence: Near-Optimal, Anytime, Adaptive Algorithm Configuration

**Robert Kleinberg**
Department of Computer Science
Cornell University
rdk@cs.cornell.edu

**Kevin Leyton-Brown**
Department of Computer Science
University of British Columbia
kevinlb@cs.ubc.ca

**Brendan Lucier**
Microsoft Research
brlucier@microsoft.com

**Devon Graham**
Department of Computer Science
University of British Columbia
drgraham@cs.ubc.ca

## Abstract

Algorithm configuration methods optimize the performance of a parameterized heuristic algorithm on a given distribution of problem instances. Recent work introduced an algorithm configuration procedure ("Structured Procrastination") that provably achieves near optimal performance with high probability and with nearly minimal runtime in the worst case. It also offers an *anytime* property: it keeps tightening its optimality guarantees the longer it is run. Unfortunately, Structured Procrastination is not *adaptive* to characteristics of the parameterized algorithm: it treats every input like the worst case. Follow-up work ("LeapsAndBounds") achieves adaptivity but trades away the anytime property. This paper introduces a new algorithm, "Structured Procrastination with Confidence", that preserves the near-optimality and anytime properties of Structured Procrastination while adding adaptivity. In particular, the new algorithm will perform dramatically faster in settings where many algorithm configurations perform poorly. We show empirically both that such settings arise frequently in practice and that the anytime property is useful for finding good configurations quickly.

## 1 Introduction

Algorithm configuration is the task of searching a space of *configurations* of a given algorithm (typically represented as joint assignments to a set of algorithm parameters) in order to find a single configuration that optimizes a performance objective on a given distribution of inputs. In this paper, we focus exclusively on the objective of minimizing average runtime. Considerable progress has recently been made on solving this problem in practice via general-purpose, heuristic techniques such as ParamILS (Hutter et al., 2007, 2009), GGA (Ansótegui et al., 2009, 2015), irace (Birattari et al., 2002; López-Ibáñez et al., 2011) and SMAC (Hutter et al., 2011a,b). Notably, in the context of this paper, all these methods are *adaptive*: they surpass their worst-case performance when presented with "easier" search problems.

Recently, algorithm configuration has also begun to attract theoretical analysis. While there is a large body of less-closely related work that we survey in Section 1.3, the first nontrivial worst-case performance guarantees for general algorithm configuration with an average runtime minimization objective were achieved by a recently introduced algorithm called *Structured Procrastination (SP)* (Kleinberg et al., 2017). This work considered a worst-case setting in which an adversary causes every deterministic choice to play out as poorly as possible, but where observations of random variables are

unbiased samples. It is straightforward to argue that, in this setting, any fixed, deterministic heuristic for searching the space of configurations can be extremely unhelpful. The work therefore focuses on obtaining candidate configurations via random sampling (rather than, e.g., following gradients or taking the advice of a response surface model). Besides its use of heuristics, SMAC also devotes half its runtime to random sampling. Any method based on random sampling will eventually encounter the optimal configuration; the crucial question is the amount of time that this will take. The key result of Kleinberg et al. (2017) is that SP is guaranteed to find a near-optimal configuration with high probability, with worst-case running time that nearly matches a lower bound on what is possible and that asymptotically dominates that of existing alternatives such as SMAC.

Unfortunately, there is a fly in the ointment: SP turns out to be impractical in many cases, taking an extremely long time to run even on inputs that existing methods find easy. At the root, the issue is that SP treats every instance like the worst case, in which it is necessary to achieve a fine-grained understanding of every configuration's runtime in order to distinguish between them. For example, if every configuration is very similar but most are not quite $\varepsilon$-optimal, subtle performance differences must be identified. SP thus runs every configuration enough times that with high probability the configuration's runtime can accurately be estimated to within a $1 + \varepsilon$ factor.

## 1.1 LEAPSANDBOUNDS and CAPSANDRUNS

Weisz et al. (2018b) introduced a new algorithm, LEAPSANDBOUNDS (LB), that improves upon Structured Procrastination in several ways. First, LB improves upon SP's worst-case performance, matching its information-theoretic lower bound on running time by eliminating a log factor. Second, LB does not require the user to specify a runtime cap that they would never be willing to exceed on any run, replacing this term in the analysis with the runtime of the optimal configuration, which is typically much smaller. Third, and most relevant to our work here, LB includes an adaptive mechanism, which takes advantage of the fact that when a configuration exhibits low variance across instances, its performance can be estimated accurately with a smaller number of samples. However, the easiest algorithm configuration problems are probably those in which a few configurations are much faster on average than all other configurations. (Empirically, many algorithm configuration instances exhibit just such non-worst-case behaviour; see our empirical investigation in the Supplementary Materials.) In such cases, it is clearly unnecessary to obtain high-precision estimates of each bad configuration's runtime; instead, we only need to separate these configurations' runtimes from that of the best alternative. LB offers no explicit mechanism for doing this. LB also has a key disadvantage when compared to SP: it is not anytime, but instead must be given fixed values of $\varepsilon$ and $\delta$. Because LB is adaptive, there is no way for a user to anticipate the amount of time that will be required to prove $(\varepsilon, \delta)$-optimality, forcing a tradeoff between the risks of wasting available compute resources and of having to terminate LB before it returns an answer.

CAPSANDRUNS (CR) is a refinement of LB that was developed concurrently with the current paper; it has not been formally published, but was presented at an ICML 2018 workshop (Weisz et al., 2018a). CR maintains all of the benefits of LB, and furthermore introduces a second adaptive mechanism that does exploit variation in configurations' mean runtimes. Like LB, it is not anytime.

## 1.2 Our Contributions

Our main contribution is a refined version of SP that maintains the anytime property while aiming to observe only as many samples as necessary to separate the runtime of each configuration from that of the best alternative. We call it "Structured Procrastination with Confidence" (SPC). SPC differs from SP in that it maintains a novel form of lower confidence bound as an indicator of the quality of a particular configuration, while SP simply uses that configuration's sample mean. The consequence is that SPC spends much less time running poorly performing configurations, as other configurations quickly appear better and receive more attention. We initialize each lower bound with a trivial value: each configuration's runtime is bounded below by the fastest possible runtime, $\kappa_0$. SPC then repeatedly evaluates the configuration that has the most promising lower bound.[1] We perform

these runs by "capping" (censoring) runs at progressively doubling multiples of $\kappa_0$. If a run does not complete, SPC "procrastinates", deferring it until it has exhausted all runs with shorter captimes. Eventually, SPC observes enough completed runs of some configuration to obtain a nontrivial upper bound on its runtime. At this point, it is able to start drawing high-probability conclusions that other configurations are worse.

Our paper is focused on a theoretical analysis of SPC. We show that it identifies an approximately optimal configuration using running time that is nearly the best possible in the worst case; however, so does SP. The key difference, and the subject of our main theorem, is that SPC also exhibits near-minimal runtime beyond the worst case, in the following sense. Define an $(\varepsilon, \delta)$-suboptimal configuration to be one whose average runtime exceeds that of the optimal configuration by a factor of more than $1 + \varepsilon$, even when the suboptimal configuration's runs are capped so that a $\delta$ fraction of them fail to finish within the time limit. A straightforward information-theoretic argument shows that in order to verify that a configuration is $(\varepsilon, \delta)$-suboptimal it is sufficient—and may also be necessary, in the worst case—to run it for $O(\varepsilon^{-2} \cdot \delta^{-1} \cdot \mathrm{OPT})$ time. The running time of SPC matches (up to logarithmic factors) the running time of a hypothetical "optimality verification procedure" that knows the identity of the optimal configuration, and for each suboptimal configuration $i$ knows a pair $(\varepsilon_i, \delta_i)$ such that $i$ is $(\varepsilon_i, \delta_i)$-suboptimal and the product $\varepsilon_i^{-2} \cdot \delta_i^{-1}$ is as small as possible.

SPC is anytime in the sense that it first identifies an $(\varepsilon, \delta)$-optimal configuration for large values of $\varepsilon$ and $\delta$ and then continues to refine these values as long as it is allowed to run. This is helpful for users who have difficulty setting these parameters up front, as already discussed. SPC's strategy for progressing iteratively through smaller and smaller values of $\varepsilon$ and $\delta$ also has another advantage: it is actually faster than starting with the "final" values of $\varepsilon$ and $\delta$ and applying them to each configuration. This is because extremely weak configurations can be dismissed cheaply based on large $(\varepsilon, \delta)$ values, instead of taking more samples to estimate their runtimes more finely.

## 1.3 Other Related Work

There is a large body of related work in the multi-armed bandits literature, which does not attack quite the same problem but does similarly leverage the "optimism in the face of uncertainty" paradigm and many tools of analysis (Lai & Robbins, 1985; Auer et al., 2002; Bubeck et al., 2012). We do not survey this work in detail as we have little to add to the extensive discussion by Kleinberg et al. (2017), but we briefly identify some dominant threads in that work. Perhaps the greatest contact between the communities has occurred in the sphere of hyperparameter optimization (Bergstra et al., 2011; Thornton et al., 2013; Li et al., 2016) and in the literature on bandits with correlated arms that scale to large experimental design settings (Kleinberg, 2006; Kleinberg et al., 2008; Chaudhuri et al., 2009; Bubeck et al., 2011; Srinivas et al., 2012; Cesa-Bianchi & Lugosi, 2012; Munos, 2014; Shahriari et al., 2016). In most of this literature, all arms have the same, fixed cost; others (Guha & Munagala, 2007; Tran-Thanh et al., 2012; Badanidiyuru et al., 2013) consider a model where costs are variable but always paid in full. (Conversely, in algorithm configuration we can stop runs that exceed a captime, yielding a potentially censored sample at bounded cost.) Some influential departures from this paradigm include Kandasamy et al. (2016), Ganchev et al. (2010), and most notably Li et al. (2016); reasons why these methods are nevertheless inappropriate for use in the algorithm configuration setting are discussed at length by Kleinberg et al. (2017).

Recent work has examined the learning-theoretic foundations of algorithm configuration, inspired in part by an influential paper of Gupta & Roughgarden (2017) that framed algorithm configuration and algorithm selection in terms of learning theory. This vein of work has not aimed at a general-purpose algorithm configuration procedure, as we do here, but has rather sought sample-efficient, special-purpose algorithms for particular classes of problems, including combinatorial partitioning problems (clustering, max-cut, etc) (Balcan et al., 2017), branching strategies in tree search (Balcan et al., 2018b), and various algorithm selection problems (Balcan et al., 2018a). Nevertheless, this vein of work takes a perspective similar to our own and demonstrates that algorithm configuration has moved decisively from being solely the province of heuristic methods to being a topic for rigorous theoretical study.

## 2 Model

We define an algorithm configuration problem by the 4-tuple $(N, \Gamma, R, \kappa_0)$, where these elements are defined as follows. $N$ is a family of (potentially randomized) algorithms, which we call *configurations* to suggest that a single piece of code instantiates each algorithm under a different parameter setting. We do not assume that different configurations exhibit any sort of performance correlations, and can so capture the case of $n$ distinct algorithms by imagining a "master algorithm" with a single, $n$-valued categorical parameter. Parameters are allowed to take continuous values: $|N|$ can be uncountable. We typically use $i$ to index configurations. $\Gamma$ is a probability distribution over input instances. When the instance distribution is given implicitly by a finite benchmark set, let $\Gamma$ be the uniform distribution over this set. We typically use $j$ to index (input instance, random seed) pairs, to which we will hereafter refer simply as instances. $R(i, j)$ is the execution time when configuration $i \in N$ is run on input instance $j$. Given some value of $\theta > 0$, we define $R(i, j, \theta) = \min\{R(i, j), \theta\}$, the runtime capped at $\theta$. $\kappa_0 > 0$ is a constant such that $R(i, j) \geq \kappa_0$ for all configurations $i$ and inputs $j$.

For any timeout threshold $\theta$, let $R_\theta(i) = \mathrm{E}_{j \sim \Gamma}[R(i, j, \theta)]$ denote the average $\theta$-capped running time of configuration $i$, over input distribution $\Gamma$. Fixing some running time $\bar{\kappa} = 2^\beta \kappa_0$ that we will never be willing to exceed, the quantity $R_{\bar{\kappa}}(i)$ corresponds to the expected running time of configuration $i$ and will be denoted simply by $R(i)$. We will write $OPT = \min_i R(i)$. Given $\epsilon > 0$, a goal is to find $i^* \in N$ such that $R(i^*) \leq (1 + \epsilon)OPT$. We also consider a relaxed objective, where the running time of $i^*$ is *capped* at some threshold value $\theta$ for some small fraction of (instance, seed) pairs $\delta$.

**Definition 2.1.** *A configuration $i^*$ is $(\epsilon, \delta)$-optimal if there exists some threshold $\theta$ such that $R_\theta(i^*) \leq (1 + \epsilon)OPT$, and $\mathrm{Pr}_{j \sim \Gamma}\left(R(i^*, j) > \theta\right) \leq \delta$. Otherwise, we say $i^*$ is $(\epsilon, \delta)$-suboptimal.*

## 3 Structured Procrastination with Confidence

In this section we present and analyze our algorithm configuration procedure, which is based on the "Structured Procrastination" principle introduced in Kleinberg et al. (2017). We call the procedure SPC (Structured Procrastination with Confidence) because, compared with the original Structured Procrastination algorithm, the main innovation is that instead of approximating the running time of each configuration by taking $\widetilde{O}(1/\varepsilon^2)$ samples for some $\varepsilon$, it approximates it using a lower confidence bound that becomes progressively tighter as the number of samples increases. We focus on the case where $N$, the set of all configurations, is finite and can be iterated over explicitly. Our main result for this case is given as Theorem 3.4. In Section 4 we extend SPC to handle large or infinite spaces of configurations where full enumeration is impossible or impractical.

### 3.1 Description of the algorithm

The algorithm is best described in terms of two components: a "thread pool" of subroutines called *configuration testers*, each tasked with testing one particular configuration, and a *scheduler* that controls the allocation of time to the different configuration testers. Because the algorithm is structured in this way, it lends itself well to parallelization, but in this section we will present and analyze it as a sequential algorithm.

Each configuration tester provides, at all times, a lower confidence bound (LCB) on the average running time of its configuration. The rule for computing the LCB will be specified below; it is designed so that (with probability tending to 1 as time goes on) the LCB is less than or equal to the true average running time. The scheduler runs a main loop whose iterations are numbered $t = 1, 2, \dots$. In each iteration $t$, it polls all of the configuration testers for their LCBs, selects the one with the minimum LCB, and passes control to that configuration tester. The loop iteration ends when the tester passes control back to the scheduler. SPC is an anytime algorithm, so the scheduler's main loop is infinite; if it is prompted to return a candidate configuration at any time, the algorithm will poll each configuration tester for its "score" (described below) and then output the configuration whose tester reported the maximum score.

The way each configuration tester $i$ operates is best visualized as follows. There is an infinite stream of i.i.d. random instances $j_1, j_2, \dots$ that the tester processes. Each of them is either *completed*, *pending* (meaning we ran the configuration on that instance at least once, but it timed out before completing), or *inactive*. An instance that is completed or pending will be called active. Configuration

tester $i$ maintains state variables $\theta_i$ and $r_i$ such that the following invariants are satisfied at all times: (1) the first $r_i$ instances in the stream are active and the rest are inactive; (2) the number of pending instances is at most $q = q(r_i, t) = 50 \log(t \log r_i)$; (3) every pending instance has been attempted with timeout $\theta_i$, and no instance has been attempted with timeout greater than $2\theta_i$. To maintain these invariants, configuration tester $i$ maintains a queue of pending instances, each with a timeout parameter representing the timeout threshold to be used the next time the configuration attempts to solve the instance. When the scheduler passes control to configuration tester $i$, it either runs the pending instance at the head of its queue (if the queue has $q(r_i, t)$ elements) or it selects an inactive instance from the head of the i.i.d. stream and runs it with timeout threshold $\theta_i$. In both cases, if the run exceeds its timeout, it is reinserted into the back of the queue with the timeout threshold doubled.

At any time, if configuration tester $i$ is asked to return a score (for the purpose of selecting a candidate optimal configuration) it simply outputs $r_i$, the number of active instances. The logic justifying this choice of score function is that the scheduler devotes more time to promising configurations than to those that appear suboptimal; furthermore, better configurations run faster on average and so complete a greater number of runs. This dual tendency of near-optimal configuration testers to be allocated a greater amount of running time and to complete a greater number of runs per unit time makes the number of active instances a strong indicator of the quality of a configuration, as we formalize in the analysis.

We must finally specify how configuration tester $i$ computes its lower confidence bound on $R(i)$; see Figure 1 for an illustration. Recall that the configuration tester has a state variable $\theta_i$ and that for every active instance $j$, the value $R(i, j, \theta_i)$ is already known because $i$ has either completed instance $j$, or it has attempted instance $j$ with timeout threshold $\theta_i$. Given some iteration of the algorithm, define $G$ to be the empirical cumulative distribution function (CDF) of $R(i, j, \theta_i)$ as $j$ ranges over all the active instances. A natural estimation of $R_{\theta_i}(i)$ would be the expectation of this empirical distribution, $\int_0^\infty (1 - G(x))dx$. Our lower bound will be the expectation of a modified CDF, found by scaling $G$ non-uniformly toward 1. To formally describe the modification we require some definitions. Here and throughout this paper, we use the notation $\log(\cdot)$ to denote the base-2 logarithm and $\ln(\cdot)$ to denote the natural logarithm. Let $\epsilon(k, r, t) = \sqrt{\frac{9 \cdot 2^k \ln(kt)}{r}}$.

---

**Algorithm 1:** Structured Procrastination w/ Confidence

---

**require :** Set $N$ of $n$ algorithm configurations
**require :** Lower bound on runtime, $\kappa_0$

*// Initializations*
1   $t := 0$
2   **for** $i \in N$ **do**
3     $C_i := $ new Configuration Tester for $i$
4     $C_i$.`Initialize()`

*// Main loop. Run until interrupted.*
5   **repeat**    *// GetLCB() returns LCB as described in the text.*
6     $i := \arg\min_{i \in N} \; C_i$.`GetLCB()`
7     $C_i$.`ExecuteStep()`
8   **until** *anytime search is interrupted*
9   **return**   $i^* = \arg\max_{i \in N} \{C_i.$`GetNumActive()`$\}$

*// Configuration Testing Controller.*
10   **Class** `ConfigurationTester()`
    **require :** Sequence $j_1, j_2, \ldots$ of instances
    **require :** Global iteration counter, $t$

    **Procedure** `Initialize()`
11
12      $r := 0, \theta := \kappa_0, q = 1$
13      $Q :=$ empty double-ended queue

    **Procedure** `ExecuteStep()`
14
15      $t := t + 1$
16      **if** $|Q| < q$ **then**    *// Replenish queue*
17        $r := r + 1$
18        $\ell := r$
19      **else**
20        Remove $(\ell, \theta')$ from head of $Q$
21        $\theta := \theta'$
22      **if** $\text{RUN}(i, j_\ell, \theta)$ *terminates in time* $\tau \leq \theta$ **then**
23        $R_{i\ell\theta} := \tau$
24      **else**
25        $R_{i\ell\theta} := \theta$
26        Insert $(\ell, 2\theta)$ at tail of $Q$
27      $q := \lceil 25 \log(t \log r) \rceil$

    **Procedure** `GetNumActive()`
28
29      **return** $r$

---

For $0 < p < 1$ let

$$\beta(p, r, t) = \begin{cases} \frac{p}{1 + \epsilon(\lfloor \log(1/p) \rfloor, r, t)} & \text{if } \epsilon(\lfloor \log(1/p) \rfloor, r, t) \leq 1/2 \\ 0 & \text{otherwise.} \end{cases} \tag{1}$$

Given a function $G : [0, \infty) \to [0, 1]$, we let

$$L(G, r, t) = \int_0^\infty \beta(1 - G(x), r, t)\, dx.$$

The configuration tester's lower confidence bound is $L(G_i, r_i, t)$, where $t$ is the current iteration, $r_i$ is the number of active instances, and $G_i$ is the empirical CDF of $R(i, j, \theta_i)$.

To interpret this definition and Equation (1), think of $p$ as the value of $1 - G(x)$ for some $x$, and $\beta(p, r, t) \leq p$ as a scaled-down version of $p$. The scaling factor we use, $(1 + \epsilon(k, r, t))$, depends on the value of $p$; specifically, it increases with $k = \lfloor \log(1/p) \rfloor$. In other words, we scale $G(x)$ more aggressively as $G(x)$ gets closer to 1. If $p$ is too small as a function of $r$ and $t$, then we give up on scaling it and instead set it all the way to $\beta(p, r, t) = 0$. To see this, note that for $k$ such that $2^{-k} \leq p < 2^{1-k}$, if $k$ is large enough then we will have that $\epsilon(k, r, t) > 1/2$ so the second case of Equation (1) applies.

We also note that $L(G_i, r_i, t)$ can be explicitly computed. Observe that $G_i(x)$ is actually a step function with at most $r_i$ steps and that $G_i(x) = 1$ for $x > \theta_i$, so the integral defining $L(G_i, r_i, t)$ is actually a finite sum that can be computed in $O(r_i)$ time, given a sorted list of the elements of $\{R(i, j, \theta_i) \mid j \text{ active}\}$. Example 3.1 illustrates the gains SPC can offer over SP.

**Example 3.1.** *Suppose that there are two configurations: one that takes $100ms$ on every input and another that takes $1000ms$. With $\kappa_0 = 1ms$, $\epsilon = 0.01$, and $\zeta = 0.1$, SP will set the initial queue size of each configuration to be at least[2] $7500$, because the queue size is initialized with a value that is at least $12\epsilon^{-2} \ln(3\beta n/\zeta)$. It will run each configuration $7500$ times with a timeout of $1ms$, then it will run each of them $7500$ times with a timeout of $2ms$, then $4ms$, and so on, until it reaches $128ms$. At that point it exceeds $100ms$, so the first configuration will solve all instances in its queue. However, for the first $2 \cdot 7500 \cdot (1 + 2 + 4 + \cdots + 64) = 1.9 \times 10^6$ milliseconds of running the algorithm—more than half an hour—essentially nothing happens: SP obtains no evidence of the superiority of the first configuration.*

*In contrast, SPC maintians more modest queue sizes, and thus runs each configuration on fewer instances before running them with a timeout of $128ms$, at which point it can distinguish between the two. During the first $5000$ iterations of SPC, the size of each configuration's instance queue is at most $400$. This is because $r_i \leq t$, and $t \leq 5000$, so $q_i \leq 25 \log(5000 \log(5000)) < 400$. Further, observe that $5000$ iterations is sufficient for SPC to attempt to run both configurations on some instance with a cutoff of $128ms$, since each configuration will first run at most $400$ instances with cutoff $1ms$, then at most $400$ instances with cutoff $2ms$, and so on. Continuing up to $64ms$, for both configurations, takes a total of $2 \cdot \log(64) \cdot 400 = 4800 < 5000$ iterations. Thus, it takes at most $2 \cdot 400 \cdot (1 + 2 + 4 + ... + 64) = 101,600$ milliseconds (less than two minutes) before SPC runs each configuration on some instance with cutoff time $128ms$. We see that SPC requires significantly less time—in this example, almost a factor of $20$ less—to reach the point where it can distinguish between the two configurations.*

### 3.2 Justification of lower confidence bound

In this section we will show that for any configuration $i$ and any iteration $t$, with probability $1 - O(t^{-5/4})$ the inequality $L(G_i, r_i, t) \leq R(i)$ holds. Let $F_i$ denote the cumulative distribution function of the running time of configuration $i$. Then $R(i) = \int_0^\infty 1 - F_i(x)\, dx$, so in order to prove that $L(G_i, r_i, t) \leq R(i)$ with high probability it suffices to prove that, with high probability, for all $x$ the inequality $\beta(1 - G_i(x), r_i, t) \leq 1 - F_i(x)$ holds. To do so we will apply a multiplicative error estimate from empirical process theory due to Wellner (1978). This error estimate can be used to derive the following error bound in our setting.

**Lemma 3.2.** *Let $x_1, \ldots, x_n$ be independent random samples from a distribution with cumulative distribution function $F$, and $G$ their empirical CDF. For $0 \leq b \leq 1$, $x \geq 0$, and $0 \leq \varepsilon \leq 1/2$*

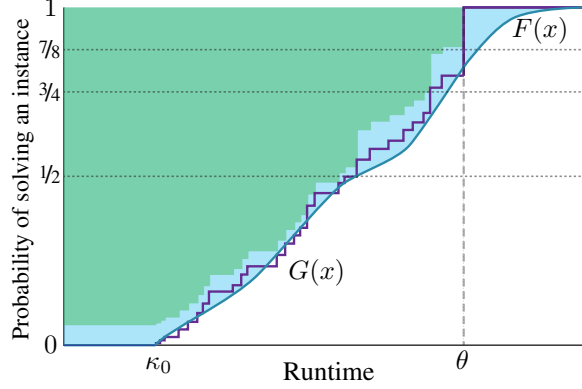

Figure 1: An illustration of how we compute the lower bound on a configuration's average runtime. The distribution of a given configuration's true runtime is $F(x)$; the empirical CDF, $G(x)$, constitutes observations sampled from $F(x)$ and censored at $\theta$. The configuration's expected runtime, the quantity we want to estimate, is the (blue) shaded region above curve $F(x)$. Our high-probability lower bound on this quantity is the (green) area above $G(x)$, scaled towards 1 as described in Equation (1).

define the events $\mathcal{E}_1(b, x) = \{1 - G(x) \geq b\}$ and $\mathcal{E}_2(\epsilon, x) = \left\{\frac{1 - G(x)}{1 + \varepsilon} > 1 - F(x)\right\}$. Then we have $\Pr\left(\exists\, x \text{ s.t. } \mathcal{E}_1(b, x) \text{ and } \mathcal{E}_2(\epsilon, x)\right) \leq \exp(-\frac{1}{4}\varepsilon^2 nb)$.

To justify the use of $L(G_i, r_i, t)$ as a lower confidence bound on $R(i)$, we apply Lemma 3.2 with $b = 2^{-k}$, $n = r$ and $\varepsilon = \varepsilon(k, r, t)$. With these parameters, $\frac{1}{4}\varepsilon^2 nb = \frac{9}{4}\ln(kt)$, hence the lemma implies the following for all $k, r, t$:

$$\Pr\left(\exists\, x \text{ s.t. } \mathcal{E}_1(2^{-k}, x) \text{ and } \mathcal{E}_2(\varepsilon(k, r, t), x)\right) \leq (kt)^{-9/4}. \tag{2}$$

The inequality is used in the following proposition to show that $L(G_i, r_i, t)$ is a lower bound on $R(i)$ with high probability.

**Lemma 3.3.** *For each configuration tester, $i$, and each loop iteration $t$,*

$$\Pr\left(\exists x \text{ s.t. } \beta(1 - G_i(x), r_i, t) > 1 - F_i(x)\right) = O(t^{-5/4}). \tag{3}$$

*Consequently* $\Pr\left(L(G_i, r_i, t) > R(i)\right) = O(t^{-5/4})$.

### 3.3 Running time analysis

Since SPC spends less time running bad configurations, we are able to show an improved runtime bound over SP. Suppose that $i$ is $(\varepsilon, \delta)$-suboptimal. We bound the expected amount of time devoted to running $i$ during the first $t$ loop iterations. We show that this quantity is $O(\varepsilon^{-2}\delta^{-1}\log(t\log(1/\delta)))$. Summing over $(\varepsilon, \delta)$-suboptimal configurations yields our main result, which is that Algorithm 1 is extremely unlikely to return an $(\epsilon, \delta)$-suboptimal configuration once its runtime exceeds the average runtime of the best configuration by a given factor. Write $B(t, \varepsilon, \delta) = \varepsilon^{-2}\delta^{-1}\log(t\log(1/\delta))$.

**Theorem 3.4.** *Fix $\varepsilon$ and $\delta$ and let $S$ be the set of $(\varepsilon, \delta)$-optimal configurations. For each $i \notin S$ suppose that $i$ is $(\varepsilon_i, \delta_i)$-suboptimal, with $\varepsilon_i \geq \varepsilon$ and $\delta_i \geq \delta$. Then if the time spent running SPC is*

$$\Omega\left(R(i^*)\left(|S| \cdot B(t, \varepsilon, \delta) + \sum_{i \notin S} B(t, \varepsilon_i, \delta_i)\right)\right),$$

*where $i^*$ denotes an optimal configuration, then SPC will return an $(\varepsilon, \delta)$-optimal configuration when it is terminated, with high probability in $t$.*

Rather than having an additive $O(\epsilon^{-2}\delta^{-1})$ term for each of $n$ configurations considered (as is the case with SP), the bound in Theorem 3.4 has a term of the form $O(\epsilon_i^{-2}\delta_i^{-1})$, for each configuration $i$ that is not $(\epsilon, \delta)$-optimal, where $\epsilon_i^{-2}\delta_i^{-1}$ is as small as possible. This can be a significant improvement in cases where many configurations being considered are far from being $(\epsilon, \delta)$-optimal. To prove Theorem 3.4, we will make use of the following lemma, which bounds the time spent running configuration $i$ in terms of its lower confidence bound and number of active instances.

**Lemma 3.5.** *At any time, if the configuration tester for configuration $i$ has $r_i$ active instances and lower confidence bound $L_i$, then the total amount of running time that has been spent running configuration $i$ is at most $9 r_i L_i$.*

The intuition is that because execution timeouts are successively doubled, the total time spent running on a given input instance $j$ is not much more than the time of the most recent execution on $j$. But if

we take an average over all active $j$, the total time spent on the most recent runs is precisely $r$ times the average runtime under the empirical CDF. The result then follows from the following lemma, Lemma 3.6, which shows that $L_i$ is at least a constant times this empirical average runtime.

**Lemma 3.6.** *At any iteration $t$, if the configuration tester for configuration $i$ has $r_i$ active instances and $G_i$ is the empirical CDF for $R(i, j, \theta_i)$, then $L(G_i, r_i, t) \geq \frac{2}{3} \int_0^{\theta_i} (1 - G_i(x)) \, dx$.*

Given Lemma 3.5, it suffices to argue that a sufficiently suboptimal configuration will have few active instances. This is captured by the following lemma.

**Lemma 3.7.** *If configuration $i$ is $(\varepsilon_i, \delta_i)$-suboptimal then at any iteration $t$, the expected number of active instances for configuration tester $i$ is bounded by $O(\varepsilon_i^{-2} \delta_i^{-1} \log(t \log(1/\delta_i)))$ and the expected amount of time spent running configuration $i$ on those instances is bounded by $O(R(i^*) \cdot \varepsilon_i^{-2} \delta_i^{-1} \log(t \log(1/\delta_i)))$ where $i^*$ denotes an optimal configuration.*

Intuitively, Lemma 3.7 follows because in order for the algorithm to select a suboptimal configuration $i$, it must be that the lower bound for $i$ is less than the lower bound for an optimal configuration. Since the lower bounds are valid with high probability, this can only happen if the lower bound for configuration $i$ is not yet very tight. Indeed, it must be significantly less than $R_\phi(i)$ for some threshold $\phi$ with $\Pr_j(R(i, j) > \phi) \geq \delta_i$. However, the lower bound cannot remain this loose for long: once the threshold $\theta$ gets large enough relative to $\phi$, and we take sufficiently many samples as a function of $\epsilon_i$ and $\delta_i$, standard concentration bounds will imply that the empirical CDF (and hence our lower bound) will approximate the true runtime distribution over the range $[0, \phi]$. Once this happens, the lower bound will exceed the average runtime of the optimal distribution, and configuration $i$ will stop receiving time from the scheduler.

Lemma 3.7 also gives us a way of determining $\epsilon$ and $\delta$ from an empirical run of SPC. If SPC returns configuration $i$ at time $t$, then by Lemma 3.7 $i$ will not be $(\epsilon, \delta)$-suboptimal for any $\epsilon$ and $\delta$ for which $r_i = \Omega(\epsilon^{-2} \delta^{-1} \log(t \log(1/\delta)))$, where $r_i$ is the number of active instances for $i$ at termination time. Thus, given a choice of $\epsilon$ and the value of $r_i$ at termination, one can solve to determine a $\delta$ for which $i$ is guaranteed to be $(\epsilon, \delta)$-optimal. See Appendix E for further details.

Given Lemma 3.7, Theorem 3.4 follows from a straightforward counting argument; see Appendix B.

## 4    Handling Many Configurations

Algorithm 1 assumes a fixed set $N$ of $n$ possible configurations. In practice, these configurations are often determined by the settings of dozens or even hundreds of parameters, some of which might have continuous domains. In these cases, it is not practical for the search procedure to take time proportional to the number of all possible configurations. However, like Structured Procrastination, the SPC procedure can be modified to handle such cases. What follows is a brief discussion; due to space constraints, the details are provided in the supplementary material.

The first idea is to sample a set $\hat{N}$ of $n$ configurations from the large (or infinite) pool, and run Algorithm 1 on the sampled set. This yields an $(\epsilon, \delta)$-optimality guarantee with respect to the best configuration in $\hat{N}$. Assuming the samples are representative, this corresponds to the top $(1/n)$'th quantile of runtimes over all configurations. We can then imagine running instances of SPC in parallel with successively doubled sample sizes, appropriately weighted, so that we make progress on estimating the top $(1/2^k)$'th quantile simultaneously for each $k$. This ultimately leads to an extension of Theorem 3.4 in which, for any $\gamma > 0$, one obtains a configuration that is $(\epsilon, \delta)$-optimal with respect to OPT$^\gamma$, the top $\gamma$-quantile of configuration runtimes. This method is anytime, and the time required for a given $\epsilon$, $\delta$, and $\gamma$ is (up to log factors) OPT$^\gamma \cdot \frac{1}{\gamma}$ times the expected minimum time needed to determine whether a randomly chosen configuration is $(\epsilon, \delta)$-suboptimal relative to OPT$^\gamma$.

## 5    Experimental Results

We experiment[3] with SPC on the benchmark set of runtimes generated by Weisz et al. (2018b) for testing LEAPSANDBOUNDS. This data consists of pre-computed runtimes for 972 configurations

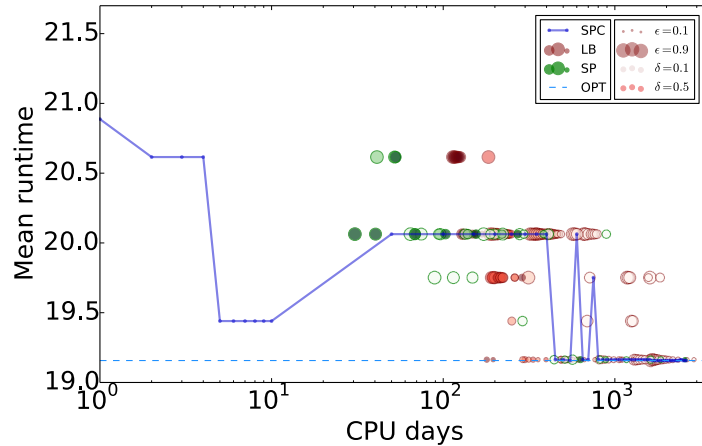

Figure 2: Mean runtimes for solutions returned by SPC after various amounts of compute time (blue line), and for those returned by LB for different $\epsilon, \delta$ pairs (red points). For LB, each point represents a different $\epsilon, \delta$ combination. Its size represents the value of $\epsilon$, and its color intensity represents the value of $\delta$. SPC is able to find a good solution relatively quickly. Different $\epsilon, \delta$ pairs can lead to drastically different runtimes, while still returning the same configuration. The $x$-axis is in log scale.

of the `minisat` (Sorensson & Een, 2005) SAT solver on 20118 SAT instances generated using CNFuzzDD[4]. A key difference between SPC and LB is the former's anytime guarantee: unlike with LB, users need not choose values of $\epsilon$ or $\delta$ in advance. Our experiments investigate the impact of this property. To avoid conflating the results with effects due to restarts and their interaction with the multiplier of $\theta$, all the times we considered were for the non-resuming simulated environment.

Figure 2 compares the solutions returned by SPC after various amounts of CPU compute time with those of LB and SP for different $\epsilon, \delta$ pairs chosen from a grid with $\epsilon \in [0.1, 0.9]$ and $\delta \in [0.1, 0.5]$. The $x$-axis measures CPU time in days, and the $y$-axis shows the expected runtime of the solution returned (capping at the dataset's max cap of $900s$). The blue line shows the result of SPC over time. The red points show the result of LB for different $\epsilon, \delta$ pairs, and the green points show this result for SP. The size of each point is proportional to $\epsilon$, while the color is proportional to $\delta$.

We draw two main conclusions from Figure 2. First, SPC was able to find a reasonable solution after a much smaller amount of compute time than LB. After only about 10 CPU days, SPC identified a configuration that was in the top 1% of all configurations in terms of max-capped runtime, while runs of LB took at least 100 CPU days for every $\epsilon, \delta$ combination we considered. Second, choosing a good $\epsilon, \delta$ combination for LB was not easy. One might expect that big, dark points would appear at shorter runtimes, while smaller, lighter ones would appear at higher runtimes. However, this was not the case. Instead, we see that different $\epsilon, \delta$ pairs led to drastically different total runtimes, often while still returning the same configuration. Conversely, SPC lets the user completely avoid this problem. It settles on a fairly good configuration after about 100 CPU days. If the user has a few hundred more CPU days to spare, they can continue to run SPC and eventually obtain the best solution reached by LB, and then to the dataset's true optimal value after about 525 CPU days. However, even at this time scale many $\epsilon, \delta$ pairs led to worse configurations being returned by LB than SPC.

## 6 Conclusion

We have presented Structured Procrastination with Confidence, an approximately optimal procedure for algorithm configuration. SPC is an anytime algorithm that uses a novel lower confidence bound to select configurations to explore, rather than a sample mean. As a result, SPC *adapts* to problem instances in which it is easier to discard poorly-performing configurations. We are thus able to show an improved runtime bound for SPC over SP, while maintaining the anytime property of SP.

We compare SPC to other configuration procedures on a simple benchmark set of SAT solver runtimes, and show that SPC's anytime property can be helpful in finding good configurations, especially early on in the search process. However, a more comprehensive empirical investigation is needed, in particular in the setting of many configurations. Such large-scale experiments will be a significant engineering challenge, and we leave this avenue to future work.

## Footnotes

[1] While both SPC and CR use confidence bounds to guide search, they take different approaches. Rather than rejecting configurations whose lower bounds get too large, SPC focuses on configurations with small lower bounds. By allocating a greater proportion of total runtime to such promising configurations we both improve the bounds for configurations about which we are more uncertain and allot more resources to configurations with relatively low mean runtimes about which we are more confident.

[2]The exact queue size depends on the number of active instances, but this bound suffices for our example.

[3]Code to reproduce experiments is available at `https://github.com/drgrhm/alg_config`

[4]http://fmv.jku.at/cnfuzzdd/

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
