[Supplementary Material · Practical_Near_Optimal_Algorithm_Configuration_supp.pdf]

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

Figure 3: Empirical runtime variation for different solvers and input distributions. For given $\delta$, each plot shows the fraction of configurations which are $(\epsilon, \delta)$-optimal for different values of $\varepsilon$; data from Hutter et al. (2014). **(top)** SPEAR SAT solver configurations on SWV for various $\delta$. **(bottom)** SPEAR on IBM instances and CPLEX MIP solver on various distributions, for fixed values of $\delta$.

## A  Runtime Variation in Practice

Unlike Structured Procrastination, SPC is designed to perform better when relatively few configurations are much faster on average than all others. It is thus worth asking whether this occurs in practice. We examined publicly available data from Hutter et al. (2014) (see `http://www.cs.ubc.ca/labs/beta/Projects/EPMs`), which studied the performance of two very different heuristic solvers (CPLEX, for mixed integer programs; and SPEAR, for satisfiability) on a total of 6 different benchmark distributions of practical problem instances; we investigate two distributions for each solver here. These observations were generated by randomly sampling from solvers' parameter spaces, just as SPC does; runs were given a captime of 300 seconds. We modified the data so that capped runs were recorded as having finished in 300 seconds (to bias against reporting variation in average runtimes across configurations).

We found a great deal of variation in average runtime across configurations; see Figure 3. Each plot corresponds to a specific value of $\delta$, and shows the CDF of the smallest value of $\epsilon$ for which each configuration remains $(\epsilon, \delta)$-optimal. The first row of this figure is based on different configurations of the SPEAR solver on SWV instances, with different figures corresponding to different $\delta$ values. Each figure's $x$-axis corresponds to $\varepsilon$ values (on a log scale); the $y$-axis reports the fraction of configurations that were $(\varepsilon, \delta)$-optimal for the given values of $\varepsilon$ and $\delta$. Observe that many configurations (between 1% and 6%) tie for being best for a range of small $\varepsilon$ values: this is because $\kappa_0 = 0.01$ in this setting, so fast configurations were often indistinguishable. This fraction grows with $\delta$: more configurations become indistinguishable when we sanitize their performance on larger fractions of instances. In the bottom row, the point in each graph where the CDF spikes upward corresponds to configurations where most instances were capped; thus, these graphs understate the true runtime variation.

What do these results mean for SPC? Consider SPEAR–SWV with $\delta = .5$. Only about 5% of configurations are optimal for $\epsilon$ less than about 100: i.e., even when capped runs are reported as having finished, 95% of configurations take at least 100 times longer than an optimal configuration. SPC will easily discard these configurations, allocating very little time to refining their estimates. Broadly, we see a similar pattern across the other solver–distribution pairs.

## B  Omitted Proofs

### B.1  Proof of Lemma 3.2

Recall the statement of the lemma. Let $x_1, \ldots, x_n$ be independent random samples from a distribution with cumulative distribution function $F$, and $G$ their empirical CDF. For $0 \leq b \leq 1$, $x \geq 0$, and $0 \leq \varepsilon \leq 1/2$ define the events $\mathcal{E}_1(b, x) = \{1 - G(x) \geq b\}$ and $\mathcal{E}_2(\epsilon, x) = \left\{ \frac{1 - G(x)}{1 + \varepsilon} > 1 - F(x) \right\}$.

Then we have
$$\Pr\left(\exists\, x \text{ s.t. } \mathcal{E}_1(b,x) \text{ and } \mathcal{E}_2(\epsilon,x)\right) \le \exp(-\tfrac{1}{4}\varepsilon^2 nb).$$

We show how this result follows directly from a bound of Wellner (Wellner, 1978). The *uniform empirical process* is the random functon $\Gamma_n : [0,1] \to [0,1]$ defined by drawing $n$ independent random samples $\xi_1,\ldots,\xi_n$ from the uniform distribution on $[0,1]$ and letting $\Gamma_n$ denote their empirical CDF, i.e. the cumulative distribution function of the uniform distribution on $\{\xi_1,\ldots,\xi_n\}$. Its left-continuous inverse $\Gamma_n^{-1}$ is defined by $\Gamma_n^{-1}(t) = \inf\{s \mid \Gamma_n(s) \ge t\}$. Lemma 2(i) of (Wellner, 1978) asserts that for all $\lambda \ge 1$ and $0 \le b \le 1$,

$$\Pr\left(\lambda \le \sup_{b \le t \le 1}\left\{\frac{t}{\Gamma_n^{-1}(t)}\right\}\right) \le \exp(-nbf(1/\lambda))$$

where $f(x) = x + \ln(1/x) - 1$. Reinterpreting this using the substitutions $t = \Gamma_n(s)$ and $\lambda = 1+\varepsilon$, and making use of the inequality $f\left(\frac{1}{1+\varepsilon}\right) \ge \frac{1}{4}\varepsilon^2$ for $0 \le \varepsilon \le 1/2$, we get

$$\Pr\left(1+\varepsilon \le \sup\left\{\frac{\Gamma_n(s)}{s}\,\middle|\,\Gamma_n(s) \ge b\right\}\right) \le \exp(-\tfrac{1}{4}\varepsilon^2 nb),$$
$$\forall\, 0 \le \varepsilon \le 1/2, 0 \le b \le 1$$

If $x_1, x_2, \ldots, x_n$ are i.i.d. samples drawn from an atomless distribution with cumulative distribution function $F$, then the numbers $F(x_1), \ldots, F(x_n)$ are independent uniformly distributed random samples $[0,1]$, as are $1 - F(x_1), \ldots, 1 - F(x_n)$. Hence if $G$ denotes the empirical CDF of the samples $x_1, \ldots, x_n$, then both of the random functions $1 - G(F^{-1}(1-s))$ and $G(F^{-1}(s))$ are uniform empirical processes. Applying Wellner's Lemma 2(i), and substituting $s = 1 - F(x)$, we obtain Lemma 3.2.

## B.2  Proof of Lemma 3.3

Recall the statement of the lemma: For each configuration tester, $i$, and each loop iteration $t$,

$$\Pr\left(\exists x \text{ s.t. } \beta(1 - G_i(x), r_i, t) > 1 - F_i(x)\right) = O(t^{-5/4}).$$

Consequently $\Pr\left(L(G_i, r_i, t) > R(i)\right) = O(t^{-5/4})$.

*Proof.* Sum inequality (2) over $k = 1, 2, \ldots$ and $r_i = 1, 2, \ldots, t$, and use the fact that $\sum_{k \ge 1} k^{-9/4} < \infty$, to deduce inequality (3). Integrate over $0 < x < \infty$ to derive the final inequality. $\square$

## B.3  Proof of Lemma 3.6

Recall the statement of the lemma: at any iteration $t$, if the configuration tester for configuration $i$ has $r$ active instances and $G$ is the empirical CDF for $R(i, j, \theta)$, then

$$L(G, r, t) \ge \frac{2}{3} \int_0^\theta (1 - G(x))\, dx.$$

*Proof.* Recalling that $L(G, r, t) = \int_0^\infty \beta(1 - G(x), r, t)\, dx$, it suffices to show that

$$\beta(1 - G(x), r, t) \ge \frac{2}{3}(1 - G(x)) \text{ for all } x \le \theta. \tag{4}$$

To see why (4) holds, note that $1 - G(\theta) = q(r,t)/r$ because $q(r,t)/r$ is the fraction of pending instances and they all have $R(i,j) \ge \theta$. Since $1 - G(x)$ is a non-increasing function of $x$, this implies that $1 - G(x) \ge q(r,t)/r$ for all $0 \le x \le \theta$.

Recalling the formula for $\beta(p, r, t)$, it is clear that (4) is equivalent to claiming that $\varepsilon(k, r, t) \le 1/2$ whenever $x \le \theta$ and $2^{-k} < 1 - G(x) \le 2^{1-k}$. Since $\varepsilon(k, r, t)$ is an increasing function of $k$, and $1 - G(x) \ge q(r,t)/r$, it suffices to prove that $\varepsilon(k, r, t) \le 1/2$ when $k = \lceil\log(r/q(r,t))\rceil$. For this value of $k$ we have $\varepsilon(k, r, t) \le \frac{1}{2}$ as desired. $\square$

## B.4 Proof of Lemma 3.5

Recall the statement of the lemma: at any time, if the configuration tester for configuration $i$ has $r$ active instances and lower confidence bound $L$, then the total amount of running time that has been spent running configuration $i$ is at most $9rL$.

*Proof.* For each active instance $j$, the total time spent running $i$ on $j$ is less than $6 \cdot R(i, j, \theta)$. This is because the doubling of timeout thresholds ensures that the time spent on all previous runs of $(i, j)$, combined, is at most twice the amount of time spent on the most recent run, which is at most $R(i, j, 2\theta)$. Hence, the time spent on $j$ is at most $3 \cdot R(i, j, 2\theta) \leq 6 \cdot R(i, j, \theta)$ Combining these bounds as $j$ ranges over active instances, the total time spent running $i$ in the first $t$ iterations satisfies

$$\text{(total time spent running } i) \leq 6r \int_0^\theta (1 - G(x)) \, dx, \tag{5}$$

since the integral represents the empirical average of $R(i, j, \theta)$ over the active instances $j$. The proof now follows from Lemma 3.6. $\qquad\square$

## B.5 Proof of Lemma 3.7

Recall the statement of the lemma: if configuration $i$ is $(\varepsilon_i, \delta_i)$-suboptimal then at any iteration $t$, the expected number of active instances for configuration tester $i$ is bounded by $O(\varepsilon_i^{-2}\delta_i^{-1} \log(t \log(1/\delta_i)))$ and the expected amount of time spent running configuration $i$ on those instances is bounded by $O(R(i^*) \cdot \varepsilon_i^{-2}\delta_i^{-1} \log(t \log(1/\delta_i)))$ where $i^*$ denotes an optimal configuration.

*Proof.* We claim that if $i$ is $(\varepsilon, \delta)$-suboptimal, then there is a timeout threshold $\phi$ and another configuration $i^*$ such that $R_\phi(i) > (1 + \varepsilon)R(i^*)$ and $\Pr_j(R(i, j) > \phi) \geq \delta$. We prove this formally as Claim B.1 below. Fix such an $i^*$ and $\phi$, and note that we must then have $R_\phi(i) \geq \delta\phi$. In an iteration $t$ when configuration tester $i$ is chosen, let $r, \theta$ denote the internal state parameters of configuration tester $i$ and let $G$ denote its empirical CDF. Similarly, for configuration tester $i^*$ let $r^*, \theta^*$ denote the internal state parameters and $G^*$ denote the empirical CDF. There are two cases to consider. **(I)** $L(G^*, r^*, t) > R(i^*)$. Section 3.2 showed this event has probability $O(t^{-5/4})$. Summing over $t$, in expectation this case accounts for only $O(1)$ runs of configuration $i$: **(II)** $L(G^*, r^*, t) \leq R(i^*)$. In this case, since we know that $R(i^*) < (1 + \varepsilon)^{-1}R_\phi(i)$, and the scheduler's selection rule implies that $L(G, r, t) \leq L(G^*, r^*, t)$, we may conclude that $L(G, r, t) \leq (1 + \varepsilon)^{-1}R_\phi(i)$. Letting $k_0 = \lceil \log(1/\delta) \rceil$ and recalling the formula for $\varepsilon(k_0, r, t)$, we see that for $r > 72\varepsilon^{-2}\delta^{-1} \log(t \log(1/\delta))$, we have $\varepsilon(k_0, r, t) < \varepsilon/2$ and thus $\varepsilon(k, r, t) < \varepsilon/2$ for all $k \leq k_0$. This means that

$$\int_0^\phi \beta(1 - G(x), r, t) \, dx > \frac{2}{2 + \varepsilon} \int_0^\phi (1 - G(x)) \, dx.$$

If we observe that $\mathrm{E}[1 - G(x)] = 1 - F(x)$ and that $\int_0^\phi (1 - F(x)) \, dx = R_\phi(i)$, we see that $L(G, r, t)$ is an average of $r$ i.i.d. random samples – corresponding to scaled draws from the empirical distribution $G$ – each of which lies in the range $[0, \phi]$ and has expected value greater than $(1 + \varepsilon/2)^{-1}R_\phi(i)$ (but at most $R_\phi(i)$). We wish to apply a Chernoff-Hoeffding bound to argue that these samples are sufficiently concentrated around their mean. To this end, consider scaling these random variables by $\phi$, so that they lie in $[0, 1]$ and have expected value at most $R_\phi(i)/\phi \leq \delta$. Then for $\lambda \geq 1$ and $r > \lambda \cdot 72\varepsilon^{-2}\delta^{-1} \log(t \log(1/\delta))$ the probability that the empirical average is less than or equal to $(1 + \varepsilon)^{-1}R_\phi(i)$ is bounded above by $e^{-c\lambda}$ by the Chernoff-Hoeffding Bound, where $c > 0$ is a constant. (Indeed, as $(1 + \varepsilon)^{-1}R_\phi(i) \leq (1 - \varepsilon/4)(1 + \varepsilon/2)^{-1}R_\phi(i)$ for all $\epsilon \leq 1$, we can take $c$ to be any constant less than $72/(2 * 4^2)$, so in particular $c = 2$ suffices.) Hence, the expected number of values of $r$ for which $L(G, r, t) \leq (1 + \varepsilon)^{-1}R_\phi(i)$ is $O(\varepsilon^{-2}\delta^{-1} \log(t \log(1/\delta)))$.

Let $s_i = 72\varepsilon_i^{-2}\delta_i^{-1} \log(t \log(1/\delta_i))$. The analysis of Case 2 above shows that for $r \geq s_i$ the probability that we run configuration tester $i$ at least once during the first $t$ iterations with a number of active instances equal to $r$ is at most $\exp(-cr/s_i)$. Of course, for $r < s_i$ the probability is at most 1. Summing over $r = 1, 2, \ldots$ we obtain the upper bound on the expected number of active instances at iteration $t$. The bound on combined running time is then derived using Lemma 3.5. $\qquad\square$

**Claim B.1.** *If $i$ is $(\varepsilon, \delta)$-suboptimal, then there is a timeout threshold $\phi$ and another configuration $i^*$ such that $R_\phi(i) > (1 + \varepsilon)R(i^*)$ and $\Pr_j(R(i, j) > \phi) \geq \delta$.*

*Proof.* Choose $i^*$ to be the optimal configuration with respect to uncapped runtime. By definition, a configuration $i$ is $(\varepsilon, \delta)$-suboptimal if for all $\theta$ such that $\Pr_j(R(i, j) > \theta) \leq \delta$, $R_\theta(i) > (1+\varepsilon)R(i^*)$.

Choose $\theta^* = \inf\{\theta\colon \Pr_j(R(i, j) > \theta) \leq \delta\}$. Then by continuity of $R_\theta(i)$ with respect to $\theta$, we have that $R_{\theta^*}(i) > (1 + \varepsilon)R(i^*)$ and $\Pr_j(R(i, j) > \theta) \leq \delta$, as required. □

## B.6 Proof of Theorem 3.4

Recall the statement of the theorem: fix some $\varepsilon$ and $\delta$, and let $S$ be the set of $(\varepsilon, \delta)$-optimal configurations. For each $i \notin S$ suppose that $i$ is $(\varepsilon_i, \delta_i)$-suboptimal, with $\varepsilon_i \geq \varepsilon$ and $\delta_i \geq \delta$. Then if the total time spent running SPC is

$$\Omega\left( R(i^*)\left( |S| \cdot B(t, \varepsilon, \delta) + \sum_{i \notin S} B(t, \varepsilon_i, \delta_i) \right) \right),$$

where $i^*$ denotes an optimal configuration, then SPC will return an $(\varepsilon, \delta)$-optimal configuration when it is terminated, with high probability in $t$.

*Proof.* Recall that $B(t, \varepsilon, \delta) = \varepsilon^{-2}\delta^{-1}\log(t\log(1/\delta))$. Note that $B(t, \varepsilon_i, \delta_i) \leq B(t, \varepsilon, \delta)$ for each $i \notin S$, by the choice of $\varepsilon_i$ and $\delta_i$. By Lemma 3.7, each $i \notin S$ runs for a total time of $O(R(i^*) \cdot B(t, \varepsilon_i, \delta_i))$. Thus, the configurations in $S$ together ran for a total time of at least $\Omega(R(i^*) \cdot |S| \cdot B(t, \varepsilon, \delta))$. At least one configuration $i \in S$ must therefore have run for a total time of $\Omega(R(i^*) \cdot B(t, \varepsilon, \delta))$, and hence the number of active instances for this configuration $i$ is at least $\Omega(B(t, \varepsilon, \delta))$. As this is larger than the number of active instances for each $i \notin S$, again by Lemma 3.7, we conclude that the configuration with largest number of active instances at termination time lies in $S$, as required. □

# C  Details of Handling Many Configurations

Like Structured Procrastination, the SPC procedure can be modified to handle cases where the pool of candidates is very large. Suppose we are given a (possibly infinite) pool $N$ of possible configurations, paired with an implicit probability distribution to allow sampling. One idea is to sample a set $\hat{N}$ of $n$ configurations, and then run Algorithm 1 on the sampled set. This would yield an $(\epsilon, \delta)$-optimality guarantee with respect to the best configuration in $\hat{N}$. Motivated by this idea, for any $\gamma > 0$, we will define $OPT^\gamma = \inf\{R\colon \Pr_{i \sim N}[R(i) > R] \leq \gamma\}$. That is, $OPT^\gamma$ is the top $\gamma$'th quantile of runtimes over all configurations. For a fixed $\gamma > 0$, we can sample a set $\hat{N}$ of $O(1/\gamma \cdot \log(1/\gamma))$ configurations, then run Algorithm 1 on the resulting sample. With high probability (in $1/\gamma$), the optimal configuration from $\hat{N}$, $i^*$, will have $R(i^*) < OPT^\gamma$. We then achieve a result similar to Theorem 3.4, but with $OPT^\gamma$ in place of $R(i^*)$, and with $\epsilon_i$ and $\delta_i$ now being random variables for each $i \in \hat{N}$.

This discussion assumed that we have advance knowledge of $\gamma$, but we can extend this approach to an anytime guarantee that simultaneously makes progress on every value of $\gamma$. Suppose that, instead of simply sampling a fixed number of configurations in advance, we ran many instances of SPC in parallel, one for each value of $\gamma = 2^{-1}, 2^{-2}, 2^{-3}, \ldots$. For each $k \geq 1$, we draw a sample $\hat{N}_k$ of $\Theta(k \cdot 2^k)$ configurations and execute SPC on set $\hat{N}_k$. If we share processor time in such a way that process $k$ receives a time share proportional to $1/k^2 = 1/\log(1/\gamma)^2$, then the end result is that the time required to find a configuration that is $(\epsilon, \delta)$-suboptimal with respect to $OPT^\gamma$ increases by a factor of $\log(1/\gamma)^2$, relative to the case in which $\gamma$ was given in advance. Combining these ideas, we arrive at the following extension of Theorem 3.4 for the case of large $N$. Recall that $B(t, \epsilon, \delta)$ is the runtime bound from Lemma 3.7. Given some $i \in N$ and some $\epsilon, \delta, \gamma > 0$, if $i$ is not $(\epsilon, \delta)$-optimal with respect to OPT$^\gamma$, write

$$V(i, \epsilon, \delta, \gamma, t) = \inf_{\epsilon', \delta'\colon\ i \text{ is } (\epsilon', \delta')\text{-suboptimal}} \{B(t, \epsilon', \delta')\}.$$

Otherwise, set $V(i, \epsilon, \delta, \gamma, t) = B(t, \epsilon, \delta)$. That is, $V(i, \epsilon, \delta, \gamma, t)$ is the tightest active-instance bound implied by Lemma 3.7 for configuration $i$. Write $V(\epsilon, \delta, \gamma, t) = E_{i \sim N}[V(i, \epsilon, \delta, \gamma, t)]$ for the expected number of active instances needed for a randomly sampled configuration.

**Theorem C.1.** *Choose any $\epsilon$, $\delta$, and $\gamma$. Suppose the total time $t$ spent running parallel instances of SPC, as described above, is at least $\Omega\left(OPT^\gamma \cdot \frac{\log^3(1/\gamma)}{\gamma} \cdot V(\epsilon, \delta, \gamma, t)\right)$. Then, with high probability in $t$, one of the parallel runs of SPC (corresponding to $k = \lceil \log(1/\gamma) \rceil$) will return an $(\epsilon, \delta)$-optimal configuration with respect to $OPT^\gamma$.*

We make two observations. First, Theorem C.1 must account for events where the empirical average of $V(i, \epsilon, \delta, \gamma, t)$ over sampled configurations differs significantly from its expectation, $V(\epsilon, \delta, \gamma, t)$. To bound this difference we use Wellner's theorem, as in Lemma 3.2, to show that the empirical CDF is within a constant factor of the true CDF nearly everywhere, except possibly at its lowest values (e.g., those that occur with probability at most $\gamma^{1/2}$). Even if the empirical distribution varies by a significant amount on these lowest values (up to a factor of $\gamma^{-1/2}$) this will not significantly perturb the empirical average. Second, note that the bound in Theorem C.1 is not necessarily monotone in $\gamma$, since $OPT^\gamma$ can decrease as $\gamma$ decreases. This is natural: a broader search is costly, but finding a new fastest configuration will speed up the search procedure. Thus, even if the user has a certain target value for $\gamma$ in mind, it can be strictly beneficial to allow SPC to search over smaller values of $\gamma$ as well.

# D   Details of Experiments

Figure 2 shows the mean runtime of the best configurations found by SPC after various amounts of CPU compute time, and the best configurations returned by LB for different $\epsilon, \delta$ pairs. For SPC we plot points for 1, 2, 3, 5 and 10 CPU days, as well as for every 25 CPU days from 50 to 2600. For the runs of LB, we ran all $\epsilon, \delta$ pairs, with $\epsilon$ chosen from $\{0.1, 0.15, 0.2, 0.25, \ldots, 0.9\}$, and $\delta$ chosen from $\{0.1, 0.15, 0.2, 0.25, \ldots, 0.5\}$, for a total of 153 observations. For SP we chose $\epsilon$ from $\{0.1, 0.2, \ldots, 0.9\}$, and $\delta$ from $\{0.1, 0.2, \ldots, 0.5\}$,

As in Weisz et al. (2018b), we set the $\zeta$ parameter of LB to 0.1; we used a $\theta$ multiplier of 1.25 and 2 for LB and SPC respectively. As mentioned, all the runtimes we considered were for the simulated environment, which does not allow for restarts. This is the simplest possible scenario in which we can make this comparison. However, an investigation of the effects of restarts, in particular with different values of the $\theta$ multiplier, on these algorithms is an interesting line of future work.

# E   Deriving $\epsilon$ and $\delta$ from an Empirical Execution

A run of SPC returns a configuration $i^*$. Theorem 3.4 provides an $(\epsilon, \delta)$-optimality guarantee, but we note that SPC does not explicitly report the values of $\epsilon$ and $\delta$ to the user. Indeed, an important feature of SPC is that the quality implications of Theorem 3.4 depend on the distribution of running times for the pool of configurations, so for "easy" problem instances the actual optimality guarantee attained might be significantly better than in the worst-case.

The following lemma shows that one can infer an improved runtime guarantee from the state of SPC at termination time. We make use of this approach when evaluating the performance of SPC in experiments. Roughly speaking, the configuration returned by SPC will be $(\epsilon, \delta)$-optimal when $\epsilon^2 \delta$ is inversely proportional to $r_i$, up to logarithmic factors, where recall that $r_i$ is the number of active instances for $i$.

**Lemma E.1.** *Suppose that SPC returns configuration $i^*$. Then for any $\epsilon > 0$, $\delta > 0$, and $\lambda \geq 1$ such that $\epsilon^2 \delta \geq 72\lambda \log(t \log(1/\delta))/r_i$, configuration $i^*$ is $(\epsilon, \delta)$-optimal with probability at least $1 - e^{-2\lambda}$.*

*Proof.* Suppose that SPC is terminated at time $t$. Recall from Lemma 3.7 that if a configuration $i$ is $(\epsilon, \delta)$-suboptimal, then its expected number of active instances is $O(\epsilon^{-2}\delta^{-1}\log(t \log(1/\delta)))$. Indeed, the proof of Lemma 3.7 shows something stronger: the probability that the configuration has more than $s_i = 72\epsilon^{-2}\delta^{-1}\log(t \log(1/\delta))$ active instances at time $t$ is at most $e^{-c}$ for some constant $c$, where in particular taking $c = 2$ suffices.

We conclude from this that if $r_{i^*} \geq 72\lambda\epsilon^{-2}\delta^{-1}\log(t\log(1/\delta))$, then with probability at least $1 - e^{-2\lambda}$ configuration $i^*$ is $(\epsilon, \delta)$-optimal. In other words, for any $\epsilon$ and $\delta$ such that $\epsilon^2\delta \geq 72\lambda\log(t\log(1/\delta))/r_i$, configuration $i^*$ is $(\epsilon, \delta)$-optimal with probability at least $1 - e^{-2\lambda}$. $\qquad\square$

By Lemma E.1, for any fixed $\epsilon$ we can calculate the $\delta$ for which we have an $(\epsilon, \delta)$-optimality guarantee with, e.g., probability $1/e^2$ by setting $\lambda = 1$. We also note that, up to a constant and a factor of $\log\log(1/\delta)$, this calculation corresponds to the fraction $q_i/r_i$ of pending input instances in the execution of configuration $i^*$ at termination time.