[Reviews · NeurIPS 2019]

Reviewer 1



Originality: The paper provides a novel extension of the Structured Procrastination (SP) algorithm together with a novel theoretical analysis. The algorithm seems to be a clear improvement over previous work. Quality: The authors describe and analyze their approach in detail. I would have like to see Figure 2 discussed in more detail, as it is the main empirical evaluation of the approach. I was surprised to see the deep dip around 10 CPU days that is not reached again until much later. Is there an explanation or interpretation of this? Clarity: The paper is written clearly and the algorithm, though somewhat complex, is described in approachable terms. Significance: The proposed algorithm improves over existing work in that it is both anytime and adaptive. In contrast to LB, it does not require specifying additional parameters, which makes the algorithm more practical and easier to use. This looks like very solid work, but I'm not very familiar with the area and did not attempt to follow the proofs. I have read the author response.

Reviewer 2



This is a theoretical paper extending Structured Procrastination to make it anytime. The paper is easy to follow, but I'm unfamiliar with the related work that it builds on and therefore I'm ill equipped to judge the merits of the paper in the context of existing work (e.g., novelty of the proof techniques).

Reviewer 3



This paper studies provable guarantees for algorithm configuration. Let’s say you need to solve a series of closely related computational problems that are drawn i.i.d. from an unknown distribution (e.g., SAT instances from some specific application domain). You have a number of different algorithms – or “configurations” -- you could use for this task (potentially infinitely-many), and you want to figure out which configuration has the smallest expected runtime, where the expectation is over the unknown distribution. In the past few years, three papers with provable guarantees have come out on this topic, one by Kleinberg et al. [2017] and two by Weisz et al. [2018, 2019]. This paper builds directly on the former, which introduced an algorithm called Structured Procrastination (SP). The authors of the submission call their new algorithm Structured Procrastination with Confidence (SPC). To describe the guarantees that come with SPC, let OPT be the smallest expected running time over all configurations. Given a configuration i and a problem instance j, let R(i, j) be the time it takes to solve the instance j using the configuration i (this might be infinite). Given a runtime cap t, let R(i, j, t) = min{t, R(i, j)}. A configuration is (epsilon, delta)-optimal if there exists a runtime cap t such that for all but a delta fraction of the problem instances R(i, j) <= t, and E[R(i, j, t)] <= (1 + epsilon) * OPT. One of the main differences between this work (as well as SP) and the work by Weisz et al. [2018, 2019] is that the former provides an "anytime algorithm" but the latter do not. The longer you run SP and SPC, the algorithms return configurations that are (epsilon, delta)-optimal for better and better values of epsilon and delta. Meanwhile, the work by Weisz et al. [2018, 2019] require epsilon and delta as input. The authors write that the main issue with SP is that it runs each configuration for long enough to accurately estimate its runtime to within a (1 + epsilon) factor. However, all you really need is to be able to separate the good configurations (which are (epsilon, delta)-optimal) from the bad configurations (which are not (epsilon, delta)-optimal). The authors identify the bad configurations via a new lower confidence bound (bottom of page 5) on each configuration’s expected running time. This means of computing the LCB seems technically novel, though a bit hard to wrap one’s head around. The authors make a reasonable effort to explain the intuition behind the formula on page six, and Figure 1 is especially helpful. It would be great to have more concrete comparisons with SP. For example: - Can you give a simple, but concrete, example of an algorithm configuration problem where SPC is much better than SP? - How exactly do the runtime bounds for SP and SPC compare? Here's the difference, from my understanding: Let’s say that n is the number of configurations. From what I understand, SP will terminate with an (epsilon, delta)-optimal configuration given running time OPT * n / (delta * epsilon^2). This is similar to the runtime bound provided in Theorem 3.3, but instead of multiplying each of the n configurations by 1 / (delta * epsilon^2), we multiply each suboptimal configuration i by 1 / (delta_i * epsilon_i^2), where epsilon_i and delta_i are chosen such that configuration i is (epsilon_i, delta_i)-suboptimal and 1 / (delta_i * epsilon_i^2) is as small as possible. - It would be great to compare SPC against SP in the experimental section, rather that just the algorithm LeapsAndBounds [Weisz et al., 2018]. - SP takes as input a parameter epsilon, and returns both a configuration and a value delta such that the configuration is (epsilon, delta)-optimal. In contrast, SPC doesn’t take as input epsilon, and it doesn’t return any delta --- it only returns a configuration. Is there any hope that SPC could be modified to return epsilon, delta, and a configuration that is (epsilon, delta)-optimal? It would also be great to have a more robust comparison (at least in terms of the theoretical guarantees) between SPC and LeapsAndBounds [Weisz et al., 2019], which appeared in ICML 2019 (and is concurrent work). Smaller comments: - In line 7 of the pseudocode, I think it should be C_i.ExecuteStep() - In the caption of Figure 1, it would be helpful to use the same notation as the rest of the paper (e.g., G(x) instead of \hat{f}(x))

[Author Response · NeurIPS 2019]

We thank all the reviewers for their helpful feedback. A main point raised was the need for a better comparison between Structured Procrastination (SP) and Structured Procrastination with Confidence (SPC); we will dedicate more space to this. In brief: SPC uses a novel form of lower confidence bound as an indicator of the quality of a particular configuration, while SP simply uses that configuration's sample mean. The consequence is that SPC spends much less time running poorly performing configurations, as other configurations quickly appear better and receive more attention. Since SPC spends less time running bad configurations, we are also able to show an improved runtime bound for it over SP. As Reviewer 3 points out, rather than having an additive term of the form $O(\epsilon^{-2}\delta^{-1})$ for each of $n$ configurations considered, the improved bound has a term of the form $O(\epsilon_i^{-2}\delta_i^{-1})$, for each configuration $i$ that is not $(\epsilon, \delta)$-optimal, where $\epsilon_i^{-2}\delta_i^{-1}$ is as small as possible. This can be a significant improvement in runtime in cases where many configurations being considered are far from being $(\epsilon, \delta)$-optimal. Using this confidence bound in place of the mean also requires a novel proof technique which leverages the theory of empirical processes.

The following concrete example illustrates the gains SPC can offer over SP. Suppose that there are just two configurations: one that always finishes in 100 milliseconds on every problem instance and another that always takes 1000 milliseconds. Suppose furthermore that $\kappa_0$—the minimum time it potentially takes to run a configuration—is equal to one millisecond. SP, configured with approximation parameter $\epsilon = 0.01$ and failure probability $\zeta = 0.1$, will set the initial queue size of each configuration to be at least[1] 7500, because the queue size is initialized with a value that is at least $12\epsilon^{-2}\ln(3\beta n/\zeta)$, where $\beta \geq 10$ is the logarithm of the ratio of maximum to minimum potential running times, and $n = 2$ is the number of configurations. It will run each configuration 7500 times with a timeout of 1ms, then it will run each of them 7500 times with a timeout of 2ms, then 4ms, and so on, progressively doubling the timeout until it reaches 128ms. At that point it exceeds 100ms, so the first configuration will solve all of the instances in its queue. However, for the first $2 \cdot 7500 \cdot (1 + 2 + 4 + \cdots + 64) = 1.9 \times 10^6$ milliseconds of running the algorithm—more than half an hour—essentially nothing happens: SP obtains no evidence of the superiority of the first configuration.

In contrast, SPC maintains more modest queue sizes, and thus runs each configuration on fewer instances before running them with a timeout of 128ms, at which point it can distinguish between the two configurations. In our example, during the first 5000 iterations of SPC, the size of each configuration's instance queue is at most 400. This is because $r_i \leq t$, and $t \leq 5000$, so $q_i \leq 25 \log(5000 \log(5000)) < 400$. Further, observe that 5000 iterations is sufficient for SPC to attempt to run both configurations on some instance with a cutoff of 128ms, since each configuration will first run at most 400 instances with cutoff 1ms, then at most 400 instances with cutoff 2ms, and so on. Continuing up to 64ms, for both configurations, takes a total of $2 \cdot \log(64) \cdot 400 = 4800 < 5000$ iterations. Thus, it takes at most $2 \cdot 400 \cdot (1 + 2 + 4 + ... + 64) = 101,600$ milliseconds (less than two minutes) before SPC runs each configuration on some instance with cutoff time 128ms. We see that SPC requires significantly less time—in this example, almost a factor of 20 less time—to reach the point where it can distinguish between the two configurations.

We agree with Reviewer 3 that it is important for SPC to return the parameters for which its optimality guarantee holds; we will explain how to do this in the paper. In brief: it is not possible to return every $(\epsilon, \delta)$ pair for which a guarantee is given because there are infinitely many such pairs. The original SP algorithm takes $\epsilon$ as a parameter and returns the corresponding $\delta$ for which the guarantee holds; we can do the same with SPC. We have also annotated our experimental results with the $\delta$ guaranteed as a function of time; see Figure 1.

Figure 1: $\delta$ for which optimality holds, as a function of runtime.

We experimented only with the MiniSAT data of Weisz et al (2018) since this is the dataset considered by the previous literature on provably near-optimal algorithm configuration. We are, however, also eager to see the results of more comprehensive experiments, including for the case of many configurations (i.e., continuous parameters). This is no small task, requiring significant amounts of coding and compute resources. We thus leave this important step for future work. We will of course make code available to reproduce our experiments as well.

Finally, we will of course fix all typos and other minor issues identified in the reviews.

## Footnotes

[1]The exact queue size depends on the number of active instances, but this lower bound suffices for our example.


[Meta-Review · NeurIPS 2019]

Thanks for the paper submission and for addressing the questions brought up by the reviewers. We believe this is a valuable contribution to NeurIPS. The main contribution on algorithm configuration methods, in particular enhancing Structured Procrastination with a form of adaptivity is sufficiently novel and of practical/ theoretical value. As areas of improvement, we strongly recommend clarifying the differences between SP and SPC addressed in the author feedback (this was very useful for the reviewers). Please refer to the detailed feedback for additional suggestions and comments from the reviewers.